# New Insights on the Role of ß-Cyanoalanine Synthase CAS-C1 in Root Hair Elongation through Single-Cell Proteomics

**DOI:** 10.3390/plants12234055

**Published:** 2023-12-02

**Authors:** Lucía Arenas-Alfonseca, Masashi Yamada, Luis C. Romero, Irene García

**Affiliations:** 1Instituto de Bioquímica Vegetal y Fotosíntesis (IBVF), CSIC-US, Avenida Américo Vespucio 49, 41092 Seville, Spain; lucia.arenasalfonseca@unige.ch; 2Department of Biology, Duke University, Durham, NC 27708, USA; masyamad@gate.sinica.edu.tw

**Keywords:** development, signaling, root hair, hydrogen cyanide, proteomics, *S*-cyanylation, *Arabidopsis*

## Abstract

(1) Background: Root hairs are specialized structures involved in water and plant nutrient uptake. They elongate from epidermal cells following a complex developmental program. ß-cyanoalanine synthase (CAS), which is mainly involved in hydrogen cyanide (HCN) detoxification in *Arabidopsis thaliana*, plays a role in root hair elongation, as evidenced by the fact that *cas-c1* mutants show a severe defect in root hair shape. In addition to root hairs, CAS C1 is expressed in the quiescent center and meristem. (2) Methods: To identify its role in root hair formation, we conducted single-cell proteomics analysis by isolating root hair cells using Fluorescence-activated Cell Sorting (FACS) from wild-type and *cas-c1* mutants. We also analyzed the presence of *S*-cyanylation, a protein post-translational modification (PTM) mediated by HCN and affecting cysteine residues and protein activity in proteins of wild type and *cas-c1* mutants. (3) Results and Conclusions: We have found that the *cas-c1* mutation has no visible effect on quiescent center or meristem root tissue, in both control and nutrient-deprivation conditions. We have identified more than 3900 proteins in root hairs and we have found that several proteins involved in root hair development, related to the receptor kinase FERONIA signaling and DNA methylation, are modified by *S*-cyanylation.

## 1. Introduction

Root formation is one of the most important evolutionary events in the adaptation of plants to terrestrial ecosystems. Root hairs are specialized tubular structures of epidermal cells that improve the incorporation of nutrients and anchoring of the plant to the soil, as well as playing an important role in the interaction with other organisms [1].

In *Arabidopsis*, epidermal cells present a specific distribution that determines their fate: those that are between two cortical cells will become root hairs (H cells), and those in contact with the membrane of a single cortical cell will develop into non-hair (N) cells [2]. The differentiation of N or H cells is determined by a plethora of regulatory proteins, which have been extensively studied [3]. Once cell fate is determined, H cells change their shape by modifying the cell wall to allow for root hair elongation. To accomplish this, ROP (“RHO-RELATED GTPase”) proteins are recruited by the FERONIA (FER) receptor kinase and are concentrated in the root hair growth zone [4,5]. At the same time, endoplasmic reticulum and filamentous actin are concentrated in this area. In addition, the pH decreases to 4–4.5, activating expansins and allowing the relaxation of the cell wall and the formation of the initial bulge [6]. Elongation of the root hair is directed by vesicles produced by the rough endoplasmic reticulum and the Golgi complex, which release polysaccharides and cell wall glycoproteins, as well as protein synthases and membrane transporters through exocytosis at the tip of the root hair [7]. The control of membrane trafficking, therefore, is essential in the development of root hairs. RAB GTPases such as RAB-A4b (Rab GTPase Homolog A4b) play an important role in the regulation of vesicular traffic by controlling the phosphorylation state of phosphatidyl inositol (PI)-PI4P [8,9,10]. Another ROP GTPase, ROP2 (Rho-related protein from plants 2) initiates ROS production by NADPH oxidase RHD2/ATRBOHC (root hair defective 2) [11,12]. ROS promotes the entry of Ca^2+^, which activates RHD2 in a positive feedback loop [13]. The Ca^2+^ gradient at the tip of the root hair directs the growth, facilitates the fusion of the vesicles with the apical plasma membrane and contributes to the release of the vesicular content necessary for the expansion of the cell wall [14]. The organization of the cytoskeleton, hormones such as ethylene and auxin as well as nutrient availability are also essential in the formation and development of root hairs [1,15,16].

Hydrogen cyanide (HCN) is synthesized in plants as a co-product during ethylene biosynthesis but also is produced by several plant growth-promoting rhizobacteria. It acts as a biocontrol agent and influences nutrient availability [17,18]. The role of HCN as a signaling gasotransmitter involved in root hair development and the plant immune response has been widely described [19,20]. Recently, it has also been described as a signaling molecule in mammalian cells [21,22]. HCN is mainly detoxified by the ß-cyanoalanine synthase CAS-C1 which incorporates the cyanide ion into cysteine to produce ß-cyanolanine and sulfide. Maintaining low levels of HCN by CAS-C1 is required for proper root hair development [23]. CAS-C1 accumulates in root hair tips during root hair elongation. Thus, higher accumulation of HCN in *cas-c1* inhibits the action of the NADPH oxidase RHD2 in a manner independent of enzyme inactivation [24]. Root hairs of *cas-c1* are specified correctly, but they do not elongate. Genetic analysis showed that the mutant of the *SUPERCENTIPEDE* (*SCN1*) gene is epistatic to *cas-c1* and *rhd2*. HCN acts as a signaling molecule that influences root hair development in a manner independent of ROS formation and direct NADPH oxidase inhibition, and its action has been localized in the first steps of root hair elongation, between SCN1 action and RHD2-driven production of O_2_^•−^ [24].

The mechanisms that underlie HCN modulation, the mode of action and the specific targets of this molecule are areas of active research. HCN is able to react with protein disulfide bridges and produce a post-translational modification in the thiol groups of cysteines, producing *S*-cyanocysteine. Our research has shown that some *Arabidopsis* proteins are naturally *S*-cyanylated and that this modification can alter the activity of these proteins suggesting that the *S*-cyanylation represents a new mechanism of regulation of the processes in which HCN plays a signaling role [25].

Although CAS-C1 is expressed in the meristem in addition to root hairs, we show that *cas-c1* does not affect meristem formation or root hair cell fate. In order to identify the protein targets of cyanide in the root hair, we carried out a cell-type specific proteomic approach to characterize the proteins present in wild-type root hair cells and compare them to those present in *cas-c1* mutant cells. Through this analysis, we found that several proteins involved in root hair development are modified by *S*-cyanylation.

## 2. Results

### 2.1. CAS-C1 Transcription Is Cell-Type Specific

A cell-type-specific transcriptomic analysis [26] showed that *CAS-C1* is highly expressed in several root cell types, such as the quiescent center (QC, *WOX5*-tagged cells), the phloem pole pericycle (S17-tagged cells), developing cortex (*CO*_2_-tagged cells) and mature cortex (*COR*-tagged cells) (Appendix A). Also, *CAS-C1* transcription increased in the elongation zone compared to the meristematic zone and remained high in the differentiation zone (Appendix A). These results suggested a possible developmental role of CAS-C1 in the root of *Arabidopsis thaliana*.

### 2.2. CAS-C1 Is Not Involved in QC Formation, Meristem Length or Root Elongation

To determine if *cas-c1* mutants are affected in QC establishment or maintenance, we carried out confocal microscopy of wild-type and *cas-c1* mutant meristems stained with propidium iodide to visualize cell walls. Figure 1 shows that no differences were found either in the number or morphology of QC cells in roots grown in control conditions. No differences were also observed in the meristem size of wild type and *cas-c1* (white arrows in Figure 2a mark the transition zone between meristematic and elongation zones). Time-lapse measurements of root length (Figure 2 and Appendix A) confirmed that root elongation was identical in both genotypes.

The CAS-C1 protein is involved in mitochondria sulfur metabolism. It detoxifies hydrogen cyanide by using as a substrate the sulfur-containing amino acid cysteine [27]. It is also indirectly involved in nitrogen recycling because the nitrogen residue in HCN can be incorporated into amino acids through the nitrilase activities of ß-cyanoalanine, the product of HCN detoxification by CAS-C1 [28]. To determine if root growth is affected by the loss of these metabolic functions, the root phenotype was analyzed in conditions of nitrogen and sulfur deficiency. Wild-type and *cas-c1* mutant plants were grown on vertical plates of MS medium and different nutrient-deprived media for 9 days and root length was measured (Figure 3 and Appendix A). No differences were observed in any condition between wild type and *cas-c1* (Appendix A). Phosphate starvation induces, among other root phenotypes, root hair elongation, involving auxin, ethylene, cytokinin and post-translational modifications [29]. Therefore, as *cas-c1* mutants show a defect in root hair formation, we included phosphate starvation in our observations. Again, no differences were found in inorganic phosphate (Pi) starvation (Figure 3, Appendix A). Taken together our results indicate that *cas-c1* does not affect root growth in control or under N, S or P nutritional stress conditions.

### 2.3. CAS-C1 Is Not Involved in Root Hair Cell Fate

Because *cas-c1* has a severe defect in root hair morphogenesis [23,24], we hypothesized that it may also have altered root hair cell fate. To visualize root hairs during development, we crossed into *cas-c1* the COBRA-like protein 9 (COBL9) promoter driving GFP, which is root hair specific [30,31]. The *cas-c1-*pCOBL9:GFP line showed a normal patterning of green fluorescence (Figure 4), both in its abnormal root hairs and in its root hair primordia, indicating that the patterning of root hair formation is not altered in this mutant. Thus, *cas-c1* affects root hairs exclusively in their elongation.

### 2.4. Proteomics of Wild-Type and Cas-C1 Root Hairs

To analyze the effect of the *cas-c1* mutation, we isolated 10^6^ root hair cells by fluorescence-activated cell sorting (FACS) from three independent replicate samples of WT pCOBL9:GFP and *cas-c1* pCOBL9:GFP roots. The extracted proteins from each sample were trypsin-digested and the peptides were analyzed by liquid chromatography–high-resolution mass spectrometry (LC-MS/MS).

In wild-type samples, we identified 3829 unique proteins at a false discovery rate below 1% (FDR < 1%), which represents almost 10% of the total *Arabidopsis* proteome (Dataset S1). Previous work identified 1387 proteins in root hair cells by a similar experimental approach using GeLC-MS/MS analysis [31] (Figure 5a), 90% of which we also identified. In protein extracts of the *cas-c1* pCOBL9:GFP roots, we identified 3972 proteins (Dataset S2), of which 3515 were common in both cell types, 310 were only identified in wild type and 457 only in *cas-c1* (Figure 5b).

### 2.5. Cas-C1 Root Hair Cells Differ from Non-Hair Cells at the Proteomic Level

To understand the stage at which *cas-c1* root hair development is blocked, we compared proteins present in *cas-c1* and in wild-type root hairs. To this end, we compared *cas-c1* and wild-type RH cells with published proteomics data of hair/non-hair (NH) epidermal cells [31]. Of the 457 proteins present only in *cas-c1* RH and the 113 specific to NH cells, only 5 proteins were coincident (Appendix A). Thus, at the protein level, *cas-c1* RH is not equivalent to NH cells. This is consistent with the *cas-c1* phenotype, which has root hairs, although they are short and unstructured. A similar result was obtained by comparing the proteome of *cas-c1* RH with the wild-type NH proteome in [32], in which another RH-specific promoter, EXP7, was fused to GFP and root hair cells isolated by FACS. Thus, these proteomic comparisons show that *cas-c1* root hairs are not similar to non-hair cells and they have their own identity. A deeper analysis of the root hair proteome was therefore undertaken.

### 2.6. Functional Classifications of Root Hair-Specific Proteins

To identify the function of CAS-C1 and explain the hairless phenotype of the mutant we addressed the functional classification of the proteins identified in wild-type and *cas-c1* root hair samples based on the information available in three gene/protein databases: Uniprot for Gene Ontology classification [33,34], MapMan nomenclature developed for plant-specific pathways and processes [35] and the Kyoto Encyclopedia of Genes and Genomes (KEGG) database for identification of overrepresented pathways [36]. Gene Ontology (GO) classification by biological processes of the identified proteins indicated that most of the proteins are involved in “cellular (GO:0009987) and metabolic (GO:0008152) processes”. This classification might be biased by the fact that these processes include the most abundant proteins, which are preferentially detected by MS, as observed in other proteomic analyses [37,38] (Appendix A). An important subset of proteins is categorized within the GO “response to stimulus” (GO:0050896), which includes ten secondary GO identifications, highlighting “response to stress” and “response to chemical” (Appendix A). GO classification of the proteins present only in wild type or *cas-c1* root hair samples, again showed that “cellular and metabolic processes” are the prevailing sets but within the GO classification “biological regulation” (GO:0065007) there were 66 and 136 proteins in wild type and *cas-c1*, respectively (Appendix A). Among the proteins in this GO subclassification (Appendix A), we identified the transcription factor ELONGATED HYPOCOTYL 5 (HY5), a positive regulator of photomorphogenesis whose absence results in increased root hair length [39], TRANSPARENT TESTA GLABRA 1 (TTG1), involved in trichome and root hair development [40], and ENHANCED ETHYLENE RESPONSE PROTEIN 5 (EER5), which functions in resetting the ethylene-signaling pathway [41]. In this subset of proteins, we also identified in *cas-c1* but not in wild type, several proteins involved in hormone signal transduction such as the brassinosteroid co-receptor BRI1-ASSOCIATED RECEPTOR KINASE 1 (BAK1) and the BRASSINOSTEROID-SIGNALING KINASE 7 (BSK7), and auxin signaling and trafficking such as the AUXIN SIGNALING F-BOX2 (AFB2) and SHORT AND SWOLLEN ROOT 1 (NDP1/SSR1), which encodes a tetraticopeptide-containing repeat localized in mitochondria involved in root development [42].

The identified proteins in both sample types were also analyzed based on their assigned functions and classified into 35 functional groups using the MapMan nomenclature [35] (Figure 6). The most abundant set corresponded to the general PROTEIN group, which included almost 20% of the total identified proteins in both cell type backgrounds with 853 and 898 elements in wild type and *cas-c1*, respectively. This bin set comprises several subsets involved in protein activation, protein synthesis and degradation, protein targeting, post-translational modification, protein folding and glycosylation (Appendix A). Taking into account the defect in the development of the root hairs in *cas-c1*, we analyzed in detail the proteins involved in developmental processes in the protein subset present only in the samples of wild type and *cas-c1*. As shown in Appendix A, 7 and 16 proteins classified in development are only present in wild type and *cas-c1*, respectively. Among the proteins that are absent in *cas-c1* root hair-like cells, we found the protein kinase TARGET OF RAPAMYCIN (TOR), which functions as a central regulator of growth, coordinating nutritional and hormonal signaling, as well as a repressor of the autophagy process [43], EMBRYO DEFECTIVE 30 (EMB30), a GDP/GTP exchange factor for small G-proteins involved in the specification of apical-basal pattern formation and essential for cell division and expansion [44], and HSP4/SABRE protein, which mediates microtubule organization and affects root hair patterning [45]. Among the proteins that are only present in *cas-c1*, we identified PESCADILLO, which plays a role in root elongation and differentiation [46], and TRANSPARENT TESTA GLABRA 1 (TTG1), also identified in the GO classification of biological regulation.

Finally, the proteins that were present only in wild-type or *cas-c1* samples were also analyzed using the KEGG Search Pathway (Appendix A) to identify overrepresented pathways or processes. Apart from the metabolic and secondary metabolite biosynthetic pathways, the Spliceosome pathway is the one that shows the highest representation of proteins in *cas-c1* root hair samples with 22 proteins (Appendix A and Appendix A). These 22 proteins include components of the small nuclear ribonucleoproteins U1, U2, U4, U5, U6 complex like SPF3 (At2g02570) and YLS8 (At5g08290); the Prp19 complex, and the Prp2 and Prp22 pre-mRNA-splicing factors RNA helicase. This is not surprising because the spliceosome is modified in root hair cells as compared to non-hair cells, and there is less intron retention in RH than in NH cells [32]. In addition, the RNA transport and mRNA surveillance pathways are also overrepresented with 12 and 9 proteins, respectively.

### 2.7. S-Cyanylation of Root Hair Proteins

HCN intracellularly produced or as an environmental toxicant can react with protein disulfide bridges and produce a post-translational modification of proteins by *S*-cyanylation of cysteine residues. This modification does not seem to be very stable and depending on the pH and environmental conditions it can be reversed, induce the cleavage of the polypeptide chain to form an iminothiazoline group or be released as SCN by ß-elimination reaction and generate dehydroalanine [47,48]. However, several cyanylated proteins have been identified by the direct analysis of peptides with a 25.0095 Da mass increase in the fragmentation spectrum [25]. Using this direct approach, we identified 25 proteins *S*-cyanylated in wild type (Table 1) and 32 in *cas-c1* plants (Table 2), 8 of which are common in both. The low number of modified proteins identified is biased by the fact that a chemical selective method to label, enrich and detect *S*-cyanylated proteins is not available even though such does exist for other post-translational modifications including phosphorylation [49], nitrosylation [50] or persulfidation [19]. Among the *S*-cyanylated proteins identified in *cas-c1* root hair samples, we observed four proteins of the methionine regeneration S-adenosylmethionine (SAM) cycle, including COBALAMIN-INDEPENDENT METHIONINE SYNTHASE (MS1), METHIONINE SYNTHASE 2 (MS2), the S-ADENOSYL-L-HOMOCYSTEINE (SAH) HYDROLASE 2 (SAHH2) and the S-ADENOSYL-L-HOMOCYSTEIN HYDROLASE 1 (SAHH1/HOG1). These proteins have been previously detected as *S*-cyanylated in *cas-c1,* which indicates that the SAM cycle is a significant target process of HCN [25]. Figure 7 shows the digestion pattern of the peptide containing the *S*-cyanylated Cys in SAHH1/HOG1. As shown in the table, a predicted ion containing Cys-CN is identified in the spectrum. Interestingly, *hog1* mutants show a root hair defective phenotype, similar to that of *cas-c1* [51].

We also found that the glycine-rich RNA binding protein 7 (GRP7) is *S*-cyanylated in the two sets of proteins. Figure 8 shows the digestion pattern of the peptide containing the *S*-cyanylated Cys in GRP7 and predicted ions containing Cys-CN are identified in the spectrum. GRP7 binds FER and is phosphorylated in the rapid alkalinization factor peptides (RALF)-FER-GRP7 complex, which is implicated in RNA alternative splicing during root hair formation [52]. Interestingly, *feronia* (*fer*) mutants show a very similar phenotype to *cas-c1* at the root hair level [4], therefore suggesting a relationship between *cas-c1*/HCN and *feronia*.

## 3. Discussion

The role of ROS, particularly O_2_^•−^ and H_2_O_2_, in root and root hair development has been extensively studied. These species exhibit different gradient distributions in the roots [53,54]. A gradient of O_2_^•−^ is essential for root hair elongation and this gradient at the root hair tip is lacking in the *cas-c1* mutant [23]. On the other hand, a transcriptomic analysis suggested that *CAS-C1* might play a role in QC, phloem or developing cortex formation or function [32]. Despite the fact that *cas-c1* plants have a higher ROS content than wild type in control conditions [55], no visible differences were detected in the QC, developing and maturation zone morphology or root length in *cas-c1* as compared to wild type. In the root maturation zone, the epidermal cells located in the H cell position emerge to form hair cells, and plant hormones, such as auxin and ethylene regulate root hair formation via ROS. Moreover, superoxide anion drives the elongation of the root hair via NADPH oxidase action [56]. However, the *cas-c1* phenotype is not related to ROS even though there is a possible direct inhibition of the NADPH oxidase RHD2/ATRBOHD by HCN [24]. Therefore, we have performed a proteomic analysis on isolated root hair cells to provide additional functional data for the CAS-C1 protein and HCN’s role in root hair development.

Previously, there has been little proteomic analysis performed on root hair cells [31,32]. In this work, 3972 total proteins were identified in total samples, and this increase in the number of identified proteins compared to previous works has been possible due to the improvement in the sensitivity and resolution of current mass spectrometers. From the total identified proteins, 457 were only present in *cas-c1* but not in the wild type. This demonstrated that the suppression of CAS-C1 activity and the consequent accumulation of HCN in the *cas-c1* mutant generates important changes in the root hair proteome [23,24]. The analysis of these 457 proteins using the KEGG database showed an over-representation of proteins involved in the spliceosome, the RNA transport and mRNA monitoring pathways. The spliceosome is a large complex formed by small nuclear ribonucleoproteins (snRNPs) that process introns in pre-mRNA to convert it into mature mRNA, a fundamental stage in gene expression in eukaryotes [57]. RNA binding proteins, RNA-dependent ATPases, RNA helicases and protein kinases are also involved in this process. Many of the spliceosome proteins, identified exclusively in *cas-c1*, participate in various developmental processes, including flowering, apical dominance, leaf and rosette size, shape and morphology [58,59]. An in-depth study of gene expression in specific root cells showed great variation in the presence of RNAs with non-coding intergenic regions that contribute to the functional specialization of the cell, demonstrating that alternative intron processing serves to regulate differentiation ([26]. Therefore, RH displays less intron retention than NH and this proteomic difference between *cas-c1* and wild-type plants is consistent with this finding.

It is noteworthy that we only detected TRANSPARENT TESTA GLABRA1 (TTG1) in *cas-c1* root hairs. This protein, together with WEREWOLF (WER) and GLABRA3 (Gl3), form a complex in non-hair cells that induces GLABRA2 (GL2) expression and controls non-root hair cell fate by the GL2-mediated suppression of a set of root hair specific transcription factors [60]. Therefore, a *ttg1* knockout mutant displays a hairy phenotype. The presence of abnormally high amounts of TTG1 in the *cas-c1* root hairs could result in a hairless phenotype. However, we demonstrate that the *cas-c1* mutation does not affect root hair cell fate as a COBL9-GFP fusion locates exclusively in hair cells and follows the wild-type alternate H-NH pattern, and COBL9 action locates downstream of TTG1 in the genetic pathway that leads to root hair specification and elongation [61]. Therefore, the high amount of TTG1 does not seem to be the main origin for the hairless phenotype of *cas-c1* mutants.

Previous genetic experiments pointed to ROPs as possible targets of cyanide action [24]. FERONIA regulates ROP-signaled root hair development [4], and, although the variation of gene expression does not necessarily produce the same variation in protein abundance, the transcriptional profile of *fer* mutant shows a significant number of mis-regulated genes common to mis-regulated proteins in *cas-c1* [4] (Appendix A). Thus, we can hypothesize that both *fer* and *cas-c1* share common elements in the root hair development process or that one acts upstream of the other. This may involve *S*-cyanylation of proteins regulated by FER in RH elongation as a mode of action of HCN. Candidates could be peroxidases, ROPs or other proteins involved in RH elongation mediated by FER. This opens an interesting focus on the interactions between *cas-c1*, *feronia* and *S*-cyanylation that will be the subject of future research. On the other hand, it has been demonstrated that the interaction between the peptide rapid alkalinization factor (RALF) and FERONIA modulates protein synthesis by the phosphorylation of a translation initiation factor [62]. Several initiation factors have been identified in samples of both wild-type and *cas-c1* root hairs. The translation initiation factor 2 is present only in *cas-c1* root hairs, as well as another putative initiation factor, thus supporting the possible relationship between FERONIA and CAS-C1 at the functional level. This is consistent with the MapMan analysis, in which approximately 20% of the genes belong to the PROTEIN group, opening also the question of whether proteins are affected at the post-translational level by HCN and its associated post-translational modification, the *S*-cyanylation [25]. Moreover, GRP7 is *S*-cyanylated in root hairs, being a protein that works together with the RALF-FER module to adjust the alternative RNA splicing in response to environmental and developmental factors [52]. Therefore, HCN signaling and root hair elongation could be coupled by the *S*-cyanylation of this RNA-binding protein and its interaction with FERONIA.

Although the number of *S*-cyanylated proteins identified in this work is low, we found a significant number of enzymes of the SAM cycle to be S-cyanylated, a relevant finding because in our previous work, we also found these proteins in *cas-c1* samples [25]. The SAM cycle is involved in DNA methylation through SAM, a methylation DNA agent. DNA methylation is a conserved epigenetic marker that regulates many developmental and stress responses and adaptations in plants, including the transmission of a stress memory that can be important in the response to pathogens. Mutant lines (*sahh1*/*hog1*) with reduced levels of the S-ADENOSYL-L-HOMOCYSTEIN HYDROLASE 1 enzyme show a deficiency in root hair development pointing to a relationship between methylation level and RH development [51]. There is substantial evidence that correlates the levels of SAM cycle components with DNA methylation changes [63,64]. In plants, inhibition of the METHIONINE SYNTHASE interferes with the immune response of the plant to pathogens. It has been demonstrated extensively that *cas-c1* plants are more resistant to biotrophic pathogens than wild-type plants [55,65]. In fact, very low concentrations of KCN (1 µM) are able to induce the plant response to pathogens and induce the *PATHOGENESIS-RELATED 1* (*PR1*) gene [65], and it has been very recently described that this fact is not exclusive to plants. In mammalian cells, very low concentrations of HCN (from nM to 1 µM) induce cellular proliferation and bioenergetics via cytochrome C oxidase stimulation [66]. Therefore, the role of HCN as a signal molecule that modulates important processes is increasingly challenging. 

## 4. Conclusions

The cellular analysis presented here shows that *cas-c1* mutation does not affect the root meristem morphology. A high-throughput proteomic analysis on root hair cells suggests that *S-*cyanylation of proteins related to the receptor kinase FERONIA signaling and DNA methylation can participate in root hair elongation. Although with the molecular tools currently available it is not possible to study in depth the effect of cyanide on the modification of proteins, the direct detection of this modification would allow us to further identify possible targets and intracellular processes regulated by HCN.

## 5. Materials and Methods

### 5.1. Plant Material and Growth Conditions

*Arabidopsis* (*Arabidopsis thaliana*) wild-type ecotype Col-0 and the *cas-c1* T-DNA insertion mutant (*cas-c1*; SALK_103855; [23]) were grown in soil or vertically positioned plates in a photoperiod of 16 h of white light (120 µmol m^−2^ s^−1^) at 20 °C and 8 h of dark at 18 °C. Seeds were surface sterilized using 50% (v/v) bleach and 0.1% Tween 20 (Merck, Darmstadt, Germany) for 15 min and then rinsed 5 times with sterile water. All seeds were plated on standard MS media (1× Murashige and Skoog salt mixture, Caisson Laboratories, Smithfield, UT, USA), 0.5 g/L MES, 1% Sucrose, and 1% Agar (Difco BD, Franklin Lake, NJ, USA) and adjusted to pH 5.7 with KOH. All plated seeds were stratified at 4 °C for 2 d before germination.

For nutrient deficiency assays, 1× Murashige and Skoog lacking sulfur (SO_4_^−^), nitrogen (NH_4_NO_3_) or phosphate (KH_2_PO_4_) (Caisson Laboratories) were used.

### 5.2. Root Meristem Visualization

For root meristem analysis, wild-type and *cas-c1* mutant plants were grown for 7 days in vertical MS medium plates. Roots were then stained with 10 mg/mL propidium iodide (PI) and observed using a 20× objective with a Zeiss (Oberkochen, Germany) LSM 880 laser scanning confocal microscope. Excitation and detection window were set as follows: excitation at 561 nm and detection at 570–650 nm.

### 5.3. Expression of pCOBL9:GFP in Roots 

The pCOBL9:GFP construct [30,31] was introduced into wild-type (Col) and *cas-c1* mutant plants. Two independent T3 lines in each background were grown for 6 days in MS media. Roots from transgenic plants were imaged using a Leica (Wetzlar, Germany) TCS SP2 spectral confocal microscope. Samples were excited using an argon ion laser at 488 nm; emission was detected between 510 and 580 nm for GFP imaging (pseudocolored green). The microscopy images were processed using Leica Confocal Software (version: LAS X).

### 5.4. Root Protoplasts Isolation

An amount of 500–1000 wild-type and *cas-c1* mutant seeds were grown for 6 days on Nitex nylon mesh MS vertical plates (100 mm × 100 mm × 15 mm). Roots were sliced and placed in a Petri dish (Falcon, Corning Inc, Corning, NY, USA, 351008) holding one 70 µm strainer (Falcon 352350) with 7 mL buffer B (cellulase 15 g L^−1^, (Sigma #C1794) and pectolyase 1 g L^−1^, (Sigma #P3026) in buffer A). Roots were incubated for 1 h at room temperature and 85 rpm. The filtered liquid was transferred to 15 mL Falcon and centrifuged for 6 min at 22 °C, 200× *g*. The supernatant was then removed and the pellet was resuspended in 300 µL of buffer A (mannitol 600 mM, MgCl_2_ 2 mM, CaCl_2_ 2 mM, MES 2 mM, KCl 10 mM and BSA 0.1% (p/v), pH 5.5). Then, the mix was filtered in a 70 and 40 µm (Falcon 352340) strainer, respectively, and the filtered cells were kept in a 5 mL polystyrene tube.

### 5.5. Fluorescent Activated Cell Sorting (FACS)

Protoplasts from wild-type pCOBL9:GFP and *cas-c1* pCOBL9:GFP transgenic plants were selected in a sorter MoFlo Astrios EQ (Beckman-Coulter, Brea, CA, USA), with a 70 µm nozzle at a rate of 5000 to 10,000 events per second and a fluid pressure of 60 psi. Protoplast of non-transgenic wild type and *cas-c1* plants were used as a negative control. GFP-positive cells were based on the negative control. They were read using a bandpass filter of 526/52. Protoplast were then collected in PBS buffer (NaCl 80 g L^−1^, KCl 2 g L^−1^, Na_2_HPO_4_ 14.4 g L^−1^ y KH_2_PO_4_ 2.4 g L^−1^, pH 7.4), frozen in liquid nitrogen and stored for protein extraction.

### 5.6. LC-MS/MS-Based Root Hair Proteomics

Protein extraction from sorted protoplast was carried out at the proteomics and metabolomics service of the Duke University School (https://medschool.duke.edu/research/research-support/service-centers/core-research-facilities/proteomics-and-metabolomics-4, accessed on 20 October 2023). Protein amount was measured by Bradford assay [67] and normalized with 50 mM NH_4_HCO_3_ pH 8.0 to 0.1–1 μg μL^−1^. RapiGest SF Surfactant (Waters) 0.2% (*v*/*v*) was added. Proteins from three independent replicate samples were reduced with DTT and alkylated with iodoacetamide for 30 min at RT. Protein digestion was carried out with trypsin (Promega) at a 1:50 ratio and incubated for 4 h. Peptides were dried in a vacuum centrifuge. Prior to LC-MS/MS analysis, samples were resuspended in TFA 1% (*v*/*v*) and acetonitrile 2% (*v*/*v*). Samples were then incubated for 2h at 60 °C and 100 rpm and centrifugated at 15,000 rpm for 5 min. Supernatant was stored for LC-MS/MS.

An amount of 1 µg of protein was analyzed in a LC-MS/MS elution gradient for 90 min in a mass spectrometer “Q Exactive HF-X Hybrid Quadrupole-Orbitrap”, (Thermo Fisher Scientific, Waltham, MA, USA). LCMS and MS/MS data were processed using the Proteome Discoverer 2.2 program (Thermo Scientific, Waltham, MA, USA) and analyzed for identification in MASCOT (Matrix Science, Columbus, OH, USA) against TAIR10. Search tolerances were 5 ppm for precursor ions and 0.02 Da for product ions using trypsin specificity with up to two missed cleavages. The presence of peptides containing Cys residues with cyanide modification was searched by 25.0095 D mass increase in the fragmentation spectrum, Unimod protein modification #438. All searched spectra were imported into Scaffold (v4.3, Proteome Software), and scoring thresholds were set to achieve a peptide false discovery rate of 1% using the PeptideProphet algorithm.

Protein function analysis and classification were performed using three biological gene/protein databases: MapMan [35], Uniprot [33,34] and the Kyoto Encyclopedia of Genes and Genomes (KEGG) [36].

## Figures and Tables

**Figure 1 plants-12-04055-f001:**
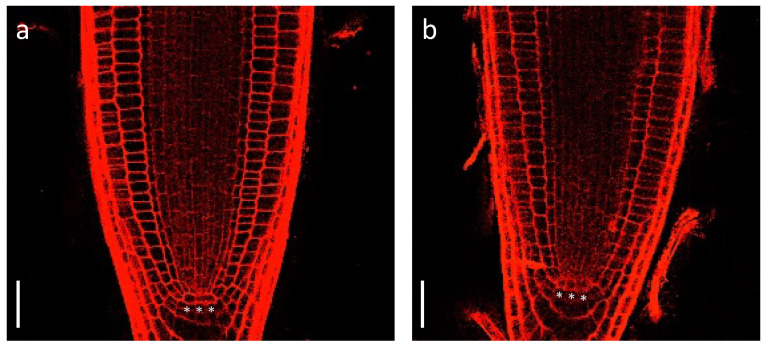
Root meristem phenotype in wild type (**a**) and *cas-c1* mutant (**b**). Seedlings were grown for 7 d on MS medium supplemented with sucrose in vertical plates. Representative images are shown. *** indicate QC cells. Propidium iodide was used to visualize cell walls. Scale bars are 200 µm.

**Figure 2 plants-12-04055-f002:**
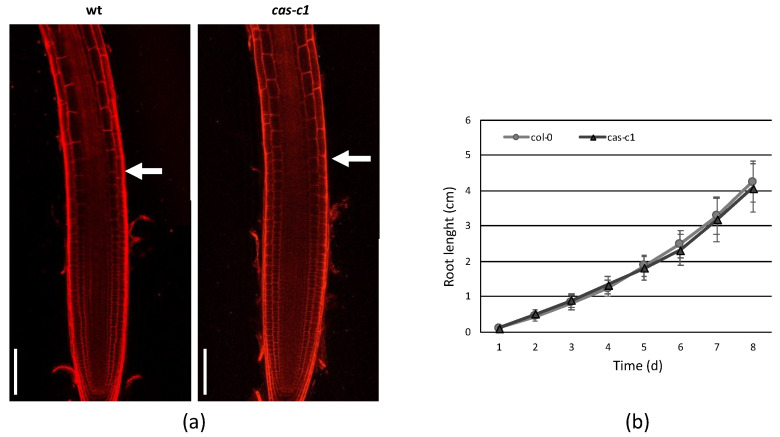
Root meristem size in MS medium. (**a**) A representative root of 7-day-old wild-type (wt) and *cas-c1* mutant was stained with propidium iodide. White arrows indicate the transition between the elongation and transition zones in wild-type and *cas-c1* mutant seedlings. Representative images are shown. (**b**) Root meristem length of wild type and *cas-c1* mutants in MS medium. Media ± SD are shown in the graph. N > 20. Scale bars are 50 µm.

**Figure 3 plants-12-04055-f003:**
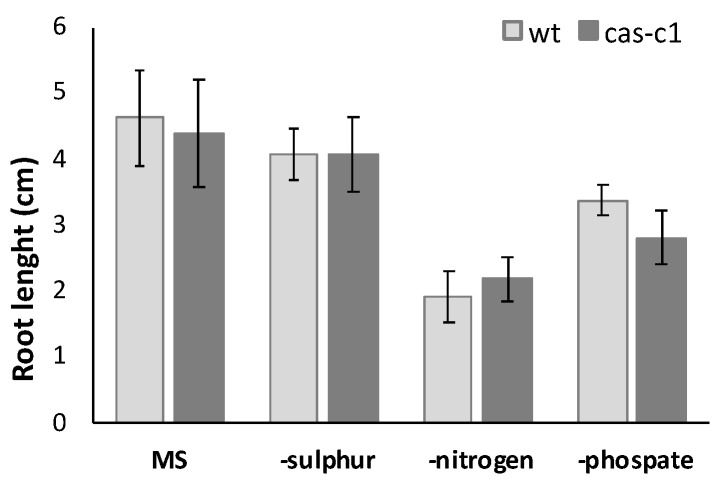
Root length measurement in nutrient deficient media. Wild-type and *cas-c1* mutant seedlings were grown for 9 d in vertical plates of MS medium or MS medium in the absence of different nutrient compounds. Root length was measured with ImageJ 1.51j8 software. Values are means ± SD of three independent experiments; ANOVA.

**Figure 4 plants-12-04055-f004:**
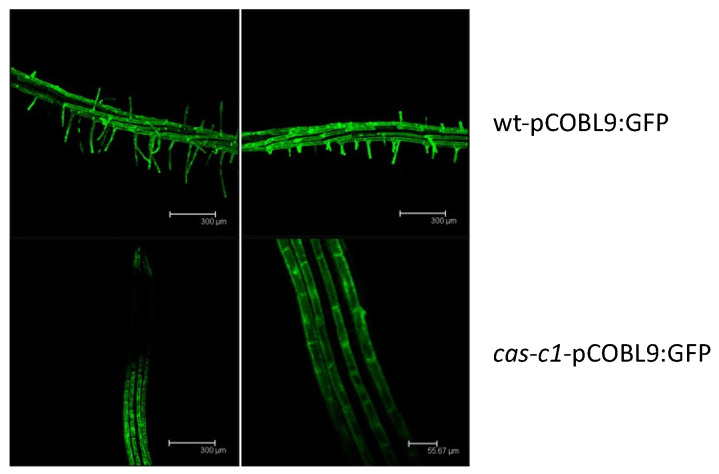
GFP localization in roots from the wt-pCOBL9:GFP and *cas-c1*-pCOBL9:GFP transgenic lines. Plants were grown for 6 days in MS medium. Representative images are shown. Scale bars are 50 μm.

**Figure 5 plants-12-04055-f005:**
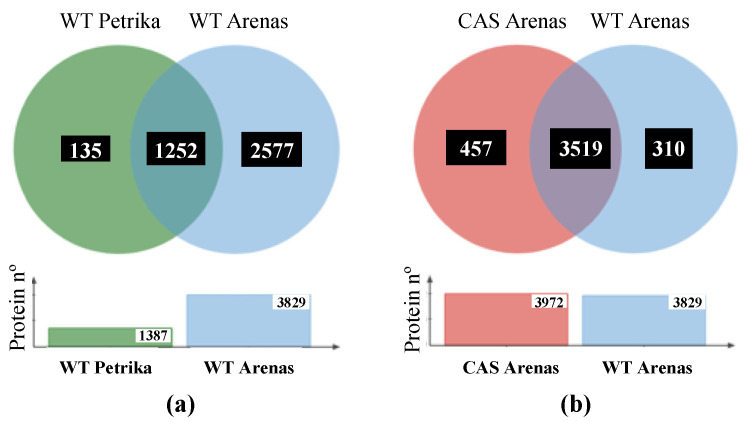
(**a**) Venn diagram showing the unique and intersection of the identified protein in the proteomic analysis of Petrika et al., 2012 and this work. (**b**) Venn diagram showing the unique and intersection of the identified proteins in wild-type and *cas-c1* root hair samples.

**Figure 6 plants-12-04055-f006:**
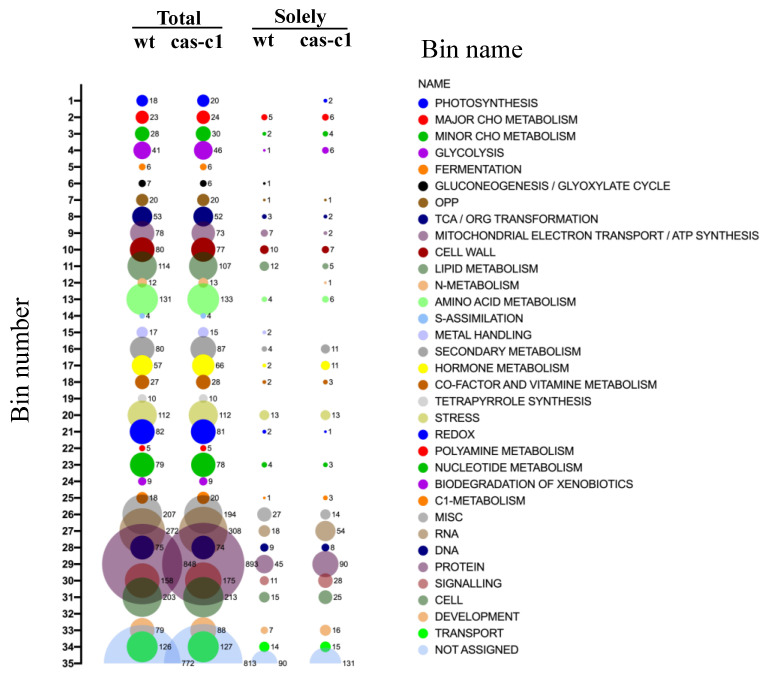
Bubble plot of the functional classification of the identified proteins according to the plant-specific database MapMan. Bubble columns represent the proteins classified within each bin of the total proteins identified in wild-type (wt), *cas-c1* root hair samples, and those only present in wt or *cas-c1* root hair samples. Numbers beside the bubbles represent the amount of proteins classified within the bin.

**Figure 7 plants-12-04055-f007:**
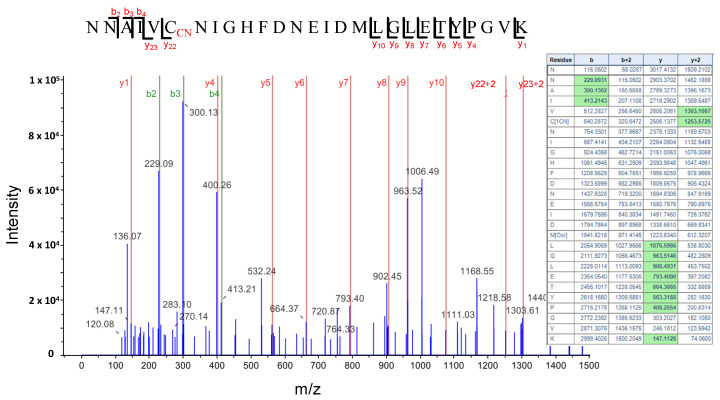
LC-MS/MS mass spectra of the tryptic peptide containing the cyanylated Cys^346^ residue of the S-Adenosyl homocysteine hydrolase SAHH1/HOG1 protein. The table contains the predicted ion type for the modified peptide, and the ions detected in the spectrum are highlighted in green color.

**Figure 8 plants-12-04055-f008:**
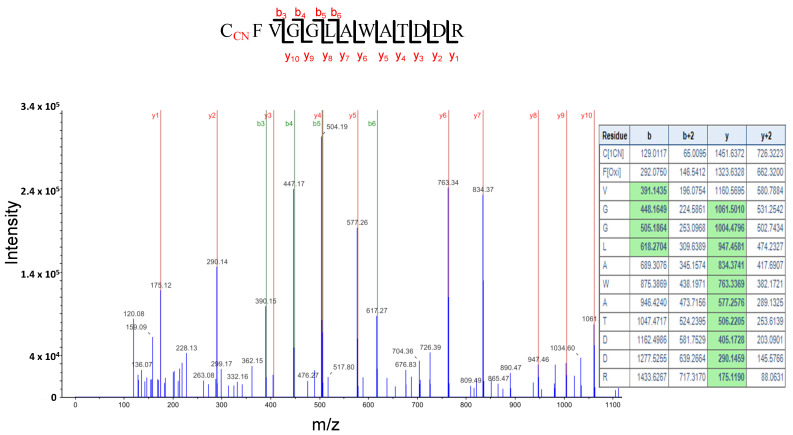
LC-MS/MS mass spectra of the tryptic peptide containing the cyanylated Cys^10^ residue of the Glycine Rich Protein 7, (GRP7). The table contains the predicted ion type for the modified peptide, and the ions detected in the spectrum are highlighted in green color.

**Table 1 plants-12-04055-t001:** Cyanylated proteins identified in wild-type root hair samples.

AGI	Gene Alias	Annotation
AT1G11910	APA1	APA1 aspartic proteinase A1
AT1G26630	ATELF5A-2FBR12	ATELF5A Eukaryotic translation initiation factor 5A-1 (eIF-5A 1) protein
AT3G52930	AtFBA8	AtFBA8 Aldolase superfamily protein
AT2G44100	ATGDI1	ATGDI1__guanosine nucleotide diphosphate dissociation inhibitor 1
AT2G21660	ATGRP7	ATGRP7 cold, circadian rhythm, and rna binding 2
AT4G39260	ATGRP8	ATGRP8 cold, circadian rhythm, and RNA binding 1
AT5G17920	ATCIMS1	ATMS1__Cobalamin-independent synthase family protein
AT3G03780	ATMS2	ATMS2 methionine synthase 2
AT5G08690		ATP synthase alpha/beta family protein
AT5G08670		ATP synthase alpha/beta family protein
AT3G26420	ATRZ-1ARBGB2	ATRZ-1A RNA-binding (RRM/RBD/RNP motifs)
AT2G30110	ATUBA1-MOS5	ATUBA1 ubiquitin-activating enzyme 1
AT5G06460	ATUBA2	ATUBA2 ubiquitin-activating enzyme 2
AT3G48340	CEP2	CEP2__Cysteine proteinases superfamily protein
AT4G24800	ECIP1-MRF3	ECIP1 MA3 domain-containing protein
AT4G34200	EDA9P-GDH1	EDA9 D-3-phosphoglycerate dehydrogenase
AT1G29880		Glycyl-tRNA synthetase/glycine--tRNA ligase
AT1G18500	IPMS1-MAML-4	IPMS1 methylthioalkylmalate synthase-like 4
AT1G53240	mMDH1	mMDH1 Lactate/malate dehydrogenase family protein
AT3G04600		Nucleotidylyl transferase superfamily protein
AT2G22780	PMDH1	PMDH1__peroxisomal NAD-malate dehydrogenase 1
AT2G29400	ATTOPP1	TOPP1__type one protein phosphatase 1
AT5G39320	UDG4	UDG4 UDP-glucose 6-dehydrogenase family protein
AT3G29360	UGD2	UGD2__UDP-glucose 6-dehydrogenase family protein
AT4G11150	VHA-E1	VHA-E1 vacuolar ATP synthase subunit E1

**Table 2 plants-12-04055-t002:** Cyanylated proteins identified in *cas-c1* root hair samples.

AGI	Gene Alias	Annotation
AT1G18080	ATARCA	ATARCA Transducin/WD40 repeat-like superfamily protein
AT2G21660	ATGRP7	ATGRP7__cold, circadian rhythm, and RNA binding 2
AT4G39260	ATGRP8	ATGRP8_cold, circadian rhythm, and RNA binding 1
AT1G27130	ATGSTU1	ATGSTU13__glutathione S-transferase tau 13
AT5G17920	ATMS1	ATMS1__Cobalamin-independent synthase family protein
AT3G03780	ATMS2	ATMS2__methionine synthase 2
AT2G20420		ATP citrate lyase (ACL) family protein
AT3G28715		ATPase, V0/A0 complex, subunit C/D
AT3G28710		ATPase, V0/A0 complex, subunit C/D
AT4G13940	ATSAHH1-HOG1	ATSAHH1__S-adenosyl-L-homocysteine hydrolase
AT3G23810	ATSAHH2	ATSAHH2__S-adenosyl-l-homocysteine (SAH) hydrolase 2
AT3G48340	CEP2	CEP2__Cysteine proteinases superfamily protein
AT2G42490	CuAOζ-zeta	CuAOζ-zeta__Copper amine oxidase family protein
AT3G53580		diaminopimelate epimerase family protein
AT1G13950	ATELF5A	ELF5A-1__eukaryotic elongation factor 5A-1
AT1G69410	ATELF5A-3	ELF5A-3__eukaryotic elongation factor 5A-3
AT1G31070	GLCNA.UT1	GLCNA.UT1__N-acetylglucosamine-1-phosphate uridylyltransferase 1
AT5G16760	AtITPK1	ITPK1__Inositol 1,3,4-trisphosphate 5/6-kinase family protein
AT5G50850	MAB1	MAB1__Transketolase family protein
AT2G33340	MAC3B	MOS4-associated complex 3B
AT4G24800	ECIP1-MRF3	MRF3__MA3 domain-containing protein
AT4G35460	ATNTRB	NTRB__NADPH-dependent thioredoxin reductase B
AT3G54960	ATPDI1	PDIL1-3__PDI-like 1-3
AT3G15000	MORF8-RIP1	RIP1__cobalt ion binding
AT4G13930	SHM4	SHM4__serine hydroxymethyltransferase 4
AT3G18060		transducin family protein/WD-40 repeat family protein
AT2G16950	ATTRN1	TRN1__transportin 1
AT5G42980	ATH3	TRXH3__thioredoxin 3
AT5G39320	UDG4	UDG4__UDP-glucose 6-dehydrogenase family protein
AT2G21270	UFD1	UFD1__ubiquitin fusion degradation 1
AT3G29360	UGD2	UGD2__UDP-glucose 6-dehydrogenase family protein

## Data Availability

Sequence data from this article can be found on the website http://www.arabidopsis.org/ (accessed on 20 October 2023) for *Arabidopsis* genes with locus identifiers provided in this study. The mass spectrometry proteomics data have been deposited in the ProteomeXchange Consortium via the PRIDE partner repository with the dataset identifier PXD028991 [68].

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
