# Peer review of "New Insights on the Role of ß-Cyanoalanine Synthase CAS-C1 in Root Hair Elongation through Single-Cell Proteomics"

_plants, 2023, doi:10.3390/plants12234055_

Round 1
Reviewer 1 Report
Comments and Suggestions for Authors
The authors in their manuscript "New insights on the role of HCN in root hair elongation through single-cell proteomics" identify through proteomic the effect of HCN on root hair elongation, the study is well designed and the data are well presented. I have the following comments
1- The authors have to include a scale bar in Figure 1
2- The authors have to include letters or symbols to describe statistical significance in Figure 3
3- Other studies indicated that root hair length is continuously increasing with decreasing N levels (Vatter et al., 2015), how you would discuss this compared to your findings
4- The study identified TARGET OF RAPAMYCIN (TOR) among the proteins absent in cas-c1 root hair-like cells, mTOR pathway is known to regulate nutrient signaling and cell cycle progression (
Comments on the Quality of English Language
Moderate editing of English language required
Author Response
The authors in their manuscript "New insights on the role of HCN in root hair elongation through single-cell proteomics" identify through proteomic the effect of HCN on root hair elongation, the study is well designed and the data are well presented. I have the following comments
- The authors have to include a scale bar in Figure 1. This has been included.
- The authors have to include letters or symbols to describe statistical significance in Figure 3. There were no statistical differences between samples and this has been indicated, as well as the test used.
- Other studies indicated that root hair length is continuously increasing with decreasing N levels (Vatter et al., 2015), how you would discuss this compared to your findings.
Vatter et al found that in N-deprived media (5 µM), root hairs are longer than in N-sufficient (5 mM) media. HCN can eventually serve as N source, but augmenting levels of cyanide in less than 1 µM should not predictably vary nitrogen nutrition. This is clearly seen in Fig. 3, where cas-c1 mutants, with around 30-40% higher cyanide levels than wild type plants, do not show any growth difference in nitrogen-deprived media.
- The study identified TARGET OF RAPAMYCIN (TOR) among the proteins absent in cas-c1 root hair-like cells, mTOR pathway is known to regulate nutrient signaling and cell cycle progression (https://doi.org/10.110/2021.05.06.442919) through phosphorylating downstream targets, I was expecting in the mTOR kinome not the protein level.
This is an interesting remark. Nevertheless, nitrogen nutrition is not altered in cas-c1 plants as discussed before; however, TOR regulation in response to HCN is an interesting topic for a different research line.
- Moderate editing of English language required
A native English speaker has revised the manuscript; however, if you consider that this is not sufficient, we can send it to a text editing company to be corrected.
Reviewer 2 Report
Comments and Suggestions for Authors
The manuscript presents results on proteomic investigation of proteins connected to HCN modulation in root hairs of Arabidopsis plant. Wild type and mutants lacking β-cyanoalanine synthase (CAS) of Aradopsis thaliana were investigated. Proteins, potential targets of cyanide in root hairs were identified using LC-MS/MS and analysed and classified with several gene and protein databases. The manuscript is generally well organized. The introduction part contains sufficient information and the results are clearly presented, though, there are several issues that should be addressed before publication:
Figure 7 and figure 8 are mentioned and discussed in text (lines 296 and 308) but are lacking in the manuscript.
Also, the method of obtaining the results on S-cyanylated proteins should be provided.
How was the statistic for Figure 3 performed?
Hoe was the extraction of proteins for LC-MS/MS performed
Concise conclusion of the main findings should be written.
Several small issues:
Line 144. Pi should be first introduced as an abbreviation for inorganic phosphate before use
Line 291 please add „S-adenosylmethionine (SAM) cycle“
Line 277. “intracellularly”
Author Response
- Figure 7 and figure 8 are mentioned and discussed in text (lines 296 and 308) but are lacking in the manuscript. Thank you for the observation. They have been included.
- Also, the method of obtaining the results on S-cyanylated proteins should be provided. This has been included.
- How was the statistic for Figure 3 performed? The method has been indicated in the footnote.
- How was the extraction of proteins for LC-MS/MS performed.
They were extracted at the proteomics and metabolomics service of the Duke University School, and this has been indicated in the Material and Methods section.
- Concise conclusion of the main findings should be written. This has been done.
- Line 144. Pi should be first introduced as an abbreviation for inorganic phosphate before use. This has been indicated.
- Line 291 please add „S-adenosylmethionine (SAM) cycle“. This has been indicated.
- Line 277. “intracellularly” This has been corrected.
Reviewer 3 Report
Comments and Suggestions for Authors The manuscript plants-2720509-peer-review-v1entitled “New insights on the role of HCN in root hair elongation through single-cell proteomics”provided some important findings. The subject of the article is interesting and has relevance for the scientific environment of the study area; however some concerns are required to be addressed. Discussion section is well written, but further elaboration could do heaps of good for the readers of this article. The major concern regarding this study is the long-term environmental implications of this study and using hydrogen cyanide as a priming agent. Title could be modified considering ß-cyanoalanine synthase CAS instead of HCN. Keywords should be reduced to 6-7. Arabidopsis is a genus name and should be italicized. Please correct accordingly. It is advised not to start a sentence with abbreviations. Line#36: Use “play” instead of “playing”. Line#68-69 seems incomplete. Line#71: Spell check. Line#277: Revise please. Figure 3: Font is too small. Please modify the figure accordingly. Nutrient deficiencies affect the morphology of roots, e.g root diameter, angle of lateral roots, etc. Did the authors observe any such differences related to nutrient deficiencies? Line#407: Spell check. Line#435-436 need to be revised. How were the nutrient deficient media prepared? Which salts were omitted out (give details in supplementary section). Subheading 4.6 and 4.7 could be merged. Before using the abbreviation, full name should be mentioned first,e.g., SAM cycle. What purpose does it serve to compare current results with the study conducted by Petrika et al., 2012 in result section? Best wishes ! Comments on the Quality of English Language The manuscript plants-2720509-peer-review-v1entitled “New insights on the role of HCN in root hair elongation through single-cell proteomics”provided some important findings. The subject of the article is interesting and has relevance for the scientific environment of the study area; however some concerns are required to be addressed. Discussion section is well written, but further elaboration could do heaps of good for the readers of this article. The major concern regarding this study is the long-term environmental implications of this study and using hydrogen cyanide as a priming agent. Title could be modified considering ß-cyanoalanine synthase CAS instead of HCN. Keywords should be reduced to 6-7. Arabidopsis is a genus name and should be italicized. Please correct accordingly. It is advised not to start a sentence with abbreviations. Line#36: Use “play” instead of “playing”. Line#68-69 seems incomplete. Line#71: Spell check. Line#277: Revise please. Figure 3: Font is too small. Please modify the figure accordingly. Nutrient deficiencies affect the morphology of roots, e.g root diameter, angle of lateral roots, etc. Did the authors observe any such differences related to nutrient deficiencies? Line#407: Spell check. Line#435-436 need to be revised. How were the nutrient deficient media prepared? Which salts were omitted out (give details in supplementary section). Subheading 4.6 and 4.7 could be merged. Before using the abbreviation, full name should be mentioned first,e.g., SAM cycle. What purpose does it serve to compare current results with the study conducted by Petrika et al., 2012 in result section? Best wishes !Author Response
- Discussion section is well written, but further elaboration could do heaps of good for the readers of this article. The major concern regarding this study is the long-term environmental implications of this study and using hydrogen cyanide as a priming agent.
We have modified part of the discussion and added a conclusions section. The use of HCN as a priming agent is far away from the research presented here; in any case, we have softened the allusions to HCN as priming agent.
- Title could be modified considering ß-cyanoalanine synthase CAS instead of HCN. This has been modified.
- Keywords should be reduced to 6-7. Oxidative stress has been eliminated from keywords.
- Arabidopsis is a genus name and should be italicized. Please correct accordingly. This has been done.
- Line#36: Use “play” instead of “playing”. Line#68-69 seems incomplete. Line#71: Spell check. Line#277: Revise please. This has been revised.
- Figure 3: Font is too small. Please modify the figure accordingly. This has been done.
- Nutrient deficiencies affect the morphology of roots, e.g root diameter, angle of lateral roots, etc. Did the authors observe any such differences related to nutrient deficiencies? We didn’t found any morphological difference between wild type and cas-c1 mutants in any of the nutrient-deficient media shown in the manuscript.
- Line#407: Spell check. Line#435-436 need to be revised. This has been done.
- How were the nutrient deficient media prepared? Which salts were omitted out (give details in supplementary section). We have included this information in the Material and Methods section.
- Subheading 4.6 and 4.7 could be merged. This has been done.
- Before using the abbreviation, full name should be mentioned first,e.g., SAM cycle.
This has been done.
- What purpose does it serve to compare current results with the study conducted by Petrika et al., 2012 in result section?
We have compared the 2012 results with our current ones in order to check how many proteins were coincident with previous proteomic analysis, taking also into account that we identified around 250% more proteins that they did. Indeed, 90% of the genes identified in our analysis were coincident with the previous ones, therefore validating both (old and new) data.
Round 2
Reviewer 1 Report
Comments and Suggestions for Authors
I am pleased to accept the manuscript in its present form
Author Response
Dear reviewer,
Thank you for your comments.
Sincerely,
Irene García
Reviewer 2 Report
Comments and Suggestions for Authors
The Authors have addressed most of the issues from the reviewing proces.
One question still remains, authors state that they have included Figure 7 and Figure 8, that are still mentioned and discussed in text, in the manuscript but I cannot see them. Also, Figure S6 is not mentioned anywhere in the text and the Figure S10 is mentioned but is lacking in the supplementary material. Please check this.
Author Response
Dear reviewer,
Thank you for your comments. We have included figures 7 and 8 in the text, and checked and corrected the supplementary figures.
Sincerely,
Irene García
Reviewer 3 Report
Comments and Suggestions for Authors
Authors have improved the manuscript as per the comments, can be accepted in present form.
Author Response

(The authors gave the same response as above.)
